# Exploring SVA Insertion Polymorphisms in Shaping Differential Gene Expressions in the Central Nervous System

**DOI:** 10.3390/biom14030358

**Published:** 2024-03-17

**Authors:** Lauren S. Hughes, Alexander Fröhlich, Abigail L. Pfaff, Vivien J. Bubb, John P. Quinn, Sulev Kõks

**Affiliations:** 1Perron Institute for Neurological and Translational Science, Perth, WA 6009, Australia; lozshughes@gmail.com (L.S.H.); alexander.froehlich@liverpool.ac.uk (A.F.); abi.pfaff@murdoch.edu.au (A.L.P.); 2Department of Pharmacology and Therapeutics, Institute of Systems, Molecular and Integrative Biology, University of Liverpool, Liverpool L69 3BX, UK; jillbubb@liverpool.ac.uk (V.J.B.); jquinn@liverpool.ac.uk (J.P.Q.); 3Centre for Molecular Medicine and Innovative Therapeutics, Murdoch University, Perth, WA 6150, Australia

**Keywords:** SINE-VNTR-*Alu*, transposable element, whole-genome sequencing, RNA-seq, gene expression regulation, expression quantitative trait loci, amyotrophic lateral sclerosis, *HLA*, *MAPT*

## Abstract

Transposable elements (TEs) are repetitive elements which make up around 45% of the human genome. A class of TEs, known as SINE-VNTR-Alu (SVA), demonstrate the capacity to mobilise throughout the genome, resulting in SVA polymorphisms for their presence or absence within the population. Although studies have previously highlighted the involvement of TEs within neurodegenerative diseases, such as Parkinson’s disease and amyotrophic lateral sclerosis (ALS), the exact mechanism has yet to be identified. In this study, we used whole-genome sequencing and RNA sequencing data of ALS patients and healthy controls from the New York Genome Centre ALS Consortium to elucidate the influence of reference SVA elements on gene expressions genome-wide within central nervous system (CNS) tissues. To investigate this, we applied a matrix expression quantitative trait loci analysis and demonstrate that reference SVA insertion polymorphisms can significantly modulate the expression of numerous genes, preferentially in the *trans* position and in a tissue-specific manner. We also highlight that SVAs significantly regulate mitochondrial genes as well as genes within the *HLA* and *MAPT* loci, previously associated within neurodegenerative diseases. In conclusion, this study continues to bring to light the effects of polymorphic SVAs on gene regulation and further highlights the importance of TEs within disease pathology.

## 1. Introduction

Amyotrophic lateral sclerosis (ALS) is a fatal neurodegenerative disease, originally termed by the French neurologist Jean-Martin Charcot in 1869 to describe muscular atrophy (amyotrophic) and tissue scarring and the hardening of tissue within the lateral spinal cord [1,2]. ALS is the most common form of a motor neuron disease (MND) and is characterized by the progressive deterioration of both upper and lower motor neurons within the brain and spinal cord [2,3,4]. The incidence of this disease is 1.75–3 per 100,000 people per year and an increased incidence of 4–8 per 100,000 people per year in the highest ALS-risk age group (45–75 years old) [5,6]. Categorised as two groups, ALS can present as familial ALS, where at least one family member of the affected individual has ALS, accounting for up to 10% of ALS cases, or sporadic ALS (sALS), where the affected individual has no prior family history, which accounts for 90–95% of cases [7]. The clinical features of typical ALS patients consist of muscle spasticity, atrophy, muscle wasting, weakness, and death due to respiratory failure, with the average survival after symptom onset lying between 3–5 years [1,5,8,9]. 

Neurodegenerative diseases are complex disorders, involving both environmental- and genetic-factor interactions. Therefore, attempts to elucidate disease mechanisms and pathogenic genetic variants are essential. Previous research has identified more than 30 genes associated with ALS, highlighting four genes, *SOD1*, *TARDBP* (*TDP-43*), *C9ORF72*, and *FUS*, for harbouring pathogenic mutations, which cause the greatest number of ALS cases [10,11,12,13]. Although these four genes have been identified as major ALS-associated genes, a 2017 meta-analysis study demonstrated that within European and Asian populations, these genes only contribute to 47.7% and 5.2% of familial and sporadic cases, respectively [14]. Following the identification of these pathogenic mutations, several pathological mechanisms have been implicated in ALS pathogenesis including oxidative stress, mitochondrial dysfunction, axonal transport, inflammation, toxic protein aggregation, and RNA metabolism and toxicity [10]. In addition to identified genetic variants only explaining a small fraction of sporadic aetiology, twin studies have highlighted the importance of genetic risk factors within sporadic ALS, estimating a 61% heritability [15]. Therefore, as the exact causation of ALS is still undetermined, a better understanding of the disease pathogenesis and identification of genetic biomarkers is essential. To further investigate the missing heritability of ALS, Theunissen et al. proposed structural variations (SVs) as an area of potential significance [16]. SVs are classified as insertions, inversions, deletions, and microsatellites, usually of repetitive structure, that are predominantly present within non-coding DNA regions and contribute towards genomic variation [16,17]. Furthermore, SVs have demonstrated the ability to modulate gene expressions and have already been implicated in neurodegenerative diseases, such as the *C9orf72* repeat expansion in ALS and frontotemporal dementia (FTD) [18,19]. Hence, as 99% of the human genome is non-coding DNA, continued research into these regions is crucial to provide new insights into disease pathogenesis and to identify new potential targets for therapeutics. 

Repetitive DNA is a major contributor to structural genomic variation. One form of repetitive DNA is a group of endogenous transposable elements (TEs), which can exist in both a static form and a mobile form. TEs are categorised into two classes known as DNA transposons and retrotransposons, whereby the later class possesses the ability to propagate throughout the genome via a ‘copy-and-paste’ mechanism, involving an RNA intermediate [20]. This results in the insertion of a new retrotransposon copy at a new locus within the host genome. Originally dismissed as “junk” DNA, TEs are known to drive genetic diversity not only by contributing to the regulation and evolution of the genome but also by contributing to genetic instability and disease progression [21,22]. Previous research by Prudencio et al. highlighted the potential implication of retrotransposons in ALS through the analysis of repetitive-element expressions using RNA sequencing data from both healthy controls as well as *C9orf72*-expansion-positive carriers and sporadic ALS patients (*C9orf72*-negative) [23]. This research revealed that repetitive-element expressions, including retrotransposons, were significantly increased in ALS patients with the C9orf72 expansion in comparison to the *C9orf72*-negative patients and healthy controls, thus suggesting the involvement of retrotransposon in ALS [23]. However, until recently, these elements have been largely overlooked in relation to neurodegenerative diseases, even though TEs constitute around 45% of the human genome [24,25,26]. 

Retrotransposons are further subdivided into two groups, dependent on the presence of long terminal repeats (LTRs), known as LTR and non-LTR retrotransposons [21]. SINE-VNTR-*Alu* (SVA) elements are a member of the non-LTR retrotransposon family, which are typically 0.7–4 kb in length [27]. SVAs are classified by evolutionary age based on their SINE-R region into subfamily groups A–F, whereby subfamily SVA-F is the youngest in evolutionary history [28]. Full-length SVA elements contain a 5′ CT element, *Alu*-like region, GC-rich VNTR (variable-number tandem repeat), SINE (short interspersed nuclear element)-R domain and a 3′ poly-A tail [28,29,30] (Figure 1b). SVAs are major contributors to genetic diversity through a variety of different mechanisms, including acting as transcriptional regulators by providing alternative splice sites, polyadenylation signals, and promoters whilst harbouring sites for transcription factor (TF) binding, thus, modulating gene expressions [31]. SVAs are not only polymorphic in structure but ongoing mobilisation has resulted in SVAs being polymorphic for their presence or absence within the genome and thus are termed retrotransposon insertion polymorphisms (RIPs) [32]. This adds additional layers of complexity to gene expression dynamics, indicating that SVAs could be associated with a predisposition to diseases [33]. A prominent example of an SVA insertion exists within the *TAF1* locus. Previous research has demonstrated that the SVA insertion within intron 32 of this gene is associated with the onset of X-linked dystonia parkinsonism (XDP) [34]. It is suggested that this insertion results in partial intron retention and a reduction in the mRNA expression of *TAF1* [34,35]. In addition, variations within the CT element (ranging between 35 and 52 repeats) indicates an inverse correlation with the XDP age of onset [36]. 

Previous work within our group has shown that seven SVAs, polymorphic for their presence or absence, were significantly associated with Parkinson’s disease progression and differential gene expression in Parkinson’s disease (PD) patients using the Parkinson’s Progression Markers Initiative (PPMI) cohort [27,32]. This involved an SVA named SVA_67, which is located at the MAPT locus, a locus which has been implicated in neurodegenerative-disease risks including PD, FTD, and Alzheimer’s disease [37,38,39]. By using a clustered regularly interspaced short palindromic repeat (CRISPR) cell line knock-out model for SVA_67, we demonstrated that this SVA was significantly associated with the differential gene expressions of three genes at the MAPT locus, associated with neurodegeneration [40]. 

In this study, we analysed SVAs which are present in the human reference genome identified as RIPs in the cohort analysed, herein termed reference SVA RIPs, to investigate the general regulatory effects of SVAs on the brain. For this analysis, we used whole-genome sequencing (WGS) and transcriptomic data obtained from the New York Genome Centre (NYGC) ALS Consortium to elucidate the role of SVA insertion polymorphisms on gene expressions in central nervous system (CNS) tissues in healthy controls and ALS patients (Figure 1a). Our analysis demonstrated that reference SVA RIPs regulate single and multiple gene targets genome-wide within CNS tissues and display tissue-specific gene modulations by comparing spinal cord, cerebellum, motor cortex, and frontal cortex tissue analyses. Furthermore, we discovered that SVA RIPs influence the expression of genes at loci (*HLA* and *MAPT*) previously associated with ALS, highlighting that the SVA regulation of genes at this locus could be a potential mechanism involved in ALS pathology. 

## 2. Materials and Methods

### 2.1. Genotyping Reference SVAs Polymorphic for Their Presence/Absence and Disease Associations from Whole-Genome Sequencing Data from ALS Consortium Dataset

Whole-genome sequencing (WGS) data from the New York Genome Centre (NYGC), as part of the ALS Consortium dataset, were obtained in cram file format and aligned to hg38. The ALS Consortium dataset contains data from individuals diagnosed with ALS-spectrum MND; other neurological disorders (Alzheimer’s, Parkinson’s disease etc); MND and ALS with other neurological disorders; along with non-neurological controls (healthy controls). The structural variant caller Delly2 (https://github.com/dellytools/delly, accessed on 1 September 2022), with default settings, was used to genotype the ALS Consortium cohort (4403 individuals) for previously identified reference SVA sites, as outlined in [41]. In total, 92 reference SVAs polymorphic for presence/absence [41] were used for a matrix eQTL and differential gene expression analysis. The nomenclature of SVAs in this study is consistent with our previous work [27,32,42]. We had 41 gDNA samples of individuals from a Parkinson’s cohort (PPMI) that were available to PCR validate bioinformatically determined polymorphisms for the presence/absence of a specific SVA of interest (SVA_67). All 41 genotypes identified by PCR, which determined the SVA_67 genotype as homozygous present (PP) and heterozygous (PA) and homozygous absent (AA), respectively, matched with the bioinformatically generated genotypes by Delly, which validated not only the reliability of the results from our previous study but also within the cohort used in this study.

### 2.2. RNA-Seq Differential Gene Expression Analysis

To elucidate the influence of SVA RIP genotypes on differential gene expressions, RNA-seq data from CNS tissues obtained from the NYGC ALS Consortium were analysed (https://www.nygenome.org/als-consortium/, accessed on 1 September 2022). For this analysis, both ALS cases and healthy controls were combined. The Salmon quantification tool was used to quantify gene-specific expression levels from previously obtained FASTQ files (https://salmon.readthedocs.io/en/latest/, accessed on 1 September 2022). Salmon-generated quant files were imported using the tximport function in R, and raw counts were extracted using the DESeqDataSetFromTximport. The DESeq2 R package Version 1.30.0 was implemented to normalise the raw counts through the median-of-ratios method. DESeq2 was also used to detect statistically significant differential gene expressions across SVA genotype groups (AA, PA, and PP), whereby P represents SVA presence and A represents SVA absence. The results of the analysis were visualised using the ggplot2 package Version 3.5.0 in R. The Kruskal–Wallis multiple comparison test (non-parametric) was applied to determine statistical significance between genotypes. The Wilcoxon test was used for pairwise comparisons to assess significant differences between genotypes, and obtained *p* values were corrected for multiple comparisons using the false discovery rate (FDR).

### 2.3. eQTL (Expression Quantitative Trait Loci) Analysis

Matrix eQTL was applied to evaluate the genetic loci (SVA elements) regulating the expression of nearby (*cis*) and distant (*trans*, >1 Mbp away) genes, using the matrixEQTL packageVersion 2.3 in R. For this analysis, the additive linear model was applied with normalised gene expression levels, chromosomal positions of SVAs (hg38), and covariates (sex and age). Effect-size estimates were reported as beta values from the matrix eQTL. The obtained *p* values were corrected for multiple testing, and the threshold was set to 0.05 for the FDR-corrected *p* values. In our study, the beta value represents the effect size of the presence of the SVA on gene expressions. This is also known as the slope coefficient. This value is generated from the linear regression model in the matrix eQTL and highlights positive (upregulation, β > 0) and negative (downregulation, β < 0) associations, respectively.

## 3. Results

### 3.1. SVA RIPs Act as eQTL Genome-Wide in CNS Tissues

To determine the impact of reference SVA RIPs on gene expression profiles, we analysed WGS and transcriptomic data of both ALS patients (*n* = 1544) and healthy controls (*n* = 359) from the ALS Consortium cohort. For this analysis, we initially combined data from different CNS tissues, including motor cortex, frontal cortex, cerebellum, occipital cortex, temporal cortex, hippocampus, and spinal cord tissues (Figure 2). 

For the eQTL analysis, we assessed changes at a gene-based level. We demonstrate that 14,830 genomic loci were significantly differentially modulated by reference SVA insertion polymorphisms (FDR *p* < 0.05). Of these effects, 167 were *cis* regulated by SVA RIPs, whilst over 98% (14,663 targets) were *trans* regulated, where *trans* regulated is defined as genes regulated greater than 1 Mb away from the SVA site (Figure 3a). This indicates that most regulatory effects observed by these SVA RIPs impact target genes that are >1 Mb away from the SVA element itself and that these could be both direct and indirect effects (Figure 1c). In addition, single SVA elements had the capacity to regulate either a single gene or multiple gene targets. Our analysis identified that 92 reference SVA RIPs were responsible for the modulation of the 14,830 genomic loci, with SVA_16 affecting the greatest number of genes (917) (Figure 3b). Of these 92 SVA elements polymorphic for their presence or absence, 26 SVAs affect more than 200 genes in total, whilst the lowest number of targets for an SVA in this analysis was 6, indicating that all reference SVA RIPs analysed in this study had the ability to modulate expression profiles of multiple gene targets in the CNS (Figure 3b).

### 3.2. Mitochondrial Genes Are Significantly Modulated by SVA RIPs

To further investigate the capacity of reference SVA elements to act as eQTL, we analysed the beta values obtained from the eQTL analysis to determine the magnitude of the measured effects induced by SVA RIPs. The ten reference SVA RIPs with the greatest effect size on gene upregulation (highest beta value) and gene downregulation (lowest beta value), and the affected gene targets are displayed in Figure 4a,b. All top ten hits for the highest and lowest beta values are *trans*-regulatory effects. SVA_55 demonstrated the greatest increase in the activation of the *MBP* gene with a beta coefficient of 272,847, whilst SVA_70 demonstrated the greatest repressive effect of the *MT-ND1* gene with a beta coefficient of −36,441 (FDR *p* < 0.05) (Figure 4a,b). Interestingly, it was noted that only two SVAs (SVA_55 and SVA_15) were responsible for the top ten hits regarding the effect size of gene activation. Furthermore, although the top hit for the effect size of gene repressions was SVA_70, modulating the *MT-ND1* expression, our results also demonstrate that the presence of both SVA_15 and SVA_55 showed the opposite effect by upregulating the *MT-ND1* expression (Figure 4a,b). To determine the impact of SVA presence on gene modulation in more detail, we examined the top hits for SVAs with the greatest effect on gene upregulation (SVA_55) and downregulation (SVA_70) independently. For *MBP*, the presence of two copies/alleles (PP genotype) of SVA_55 significantly upregulated *MBP* gene expressions compared to the presence of one copy of SVA_55 (PA genotype) or complete absence (AA genotype) (*p* = 0.0218) (Figure 4c). In comparison to the AA and PA genotypes, individuals with the PP genotype displayed a 9.5-fold and 19.2-fold increase in *MBP* gene expressions, respectively. In contrast, individuals homozygous present for SVA_70 demonstrated a significant repression of the *MT-ND1* gene in comparison to individuals heterozygous for SVA_70, namely downregulating the gene expression of *MT-ND1* by 55% (*p* = 0.0098) (Figure 4d). No statistical significance for differential gene expressions between the PP and AA genotype was obtained for the *MT-ND1* expression using the Wilcoxon pairwise comparison with FDR-adjusted *p* values (FDR *p* < 0.05).

We next analysed the top ten most significant SVA effects based on FDR *p* values. The most significant effects observed (the lowest FDR *p* value) were SVA_67 acting on the genes *MAPK8IP1P2* and *ENSG00000285668.1*, displaying an FDR *p* value of 1.93 × 10^−303^ (Table 1). This was followed by SVA_67 acting on the gene *LRRC37A*, with an FDR *p* value of 5.12 × 10^−299^ (Table 1). Intriguingly, 6/10 hits for highest significance were *cis*-specific effects on gene upregulation by SVA_24, SVA_33, and SVA_58 and gene downregulation by SVA_67. This is different to our previous results, as the top ten reference SVAs with the greatest effect size on upregulation and downregulation were all *trans*-regulatory effects. Table 1 also shows that the four most significant *trans*-specific effects induced by SVA_15, SVA_84, SVA_87, and SVA_93 all influenced the same mitochondrial gene, *MTND4P24*, whereby SVA_84, SVA_87, and SVA_93 increased gene expressions and SVA_15 repressed gene expressions. This demonstrates that multiple SVA RIPs can act on one gene, influencing gene regulation and simultaneously activating and repressing gene expressions.

To illustrate the impact of SVA_67 on target gene expressions, we investigated the specific influence of SVA_67 allele dosage on transcriptomic profiles of the respective target genes (Figure 5). As an example, this analysis is shown for the genes *MAPK8IP1P2* and *LRRC37A*, whereby gene expressions were stratified by the SVA_67 genotype (PP, PA, and AA). An analysis using the Wilcoxon pairwise comparison with FDR-adjusted *p* values (FDR < 0.05) demonstrated that the presence of SVA_67 significantly downregulated the gene expression of both *MAPK8IP1P2* and *LRRC37A* (*p* < 0.001) (Figure 5). Significant differences were observed between all genotype groups, PP vs. PA (*MAPK8IP1P2 p* = 7.33 × 10^−237^, *LRRC37A p* = 1.95 × 10^−204^); PP vs. AA (*MAPK8IP1P2 p* = 4.26 × 10^−55^, *LRRC37A p* = 1.06 × 10^−39^); and PA vs. AA (*MAPK8IP1P2 p* = 1.33 × 10^7^, *LRRC37A p* = 2.13 × 10^−12^), for both genes (Figure 5). For individuals homozygous absent for SVA_67, a change of 262-fold and 6-fold was observed for the gene expression of *MAPK8IP1P2* and *LRRC37A*, respectively, in comparison to individuals homozygous present for SVA_67 (Figure 5).

### 3.3. SVA RIPs Demonstrate Cis Effects on HLA and MAPT Loci

To elucidate the impact of *cis*-acting SVA RIPs, we explored which SVA elements had the largest significant effects on genes regulated in *cis*, including gene upregulation (positive beta value) and downregulation (negative beta value) using beta values from the matrix eQTL analysis (FDR *p* < 0.05) (Figure 6). Here, we highlight that SVA_67 largely influences gene regulation in the *cis* of six different genes, including the upregulation of *MAPT* and *LRRC37A4P* and the downregulation of *MAPK8IP1P2*, *LRRC37A2*, *LRRC37A*, and *KANSL1* (Figure 6a). SVA_67 was responsible for the two greatest increases in the gene activation of the *MAPT* and *LRRC37A4P* genes with beta coefficients of 1384 and 1101, respectively (Figure 6a). Interestingly, six of the ten most positive hits were effects induced on *HLA* genes by SVA_24, SVA_25, SVA_27, and SVA_88 (Figure 6a). Furthermore, we demonstrate that SVA_73 had the greatest effect out of all *cis*-regulatory effects with a beta coefficient of −2843 for the gene *FCGBP*, indicating a large inhibitory effect of *FCGBP* gene expressions (Figure 6b). *FCGBP* gene expressions are further repressed by SVA_72, the second highest hit in our analysis for the top ten SVAs with the greatest effect size on gene downregulation (Figure 6b).

To determine the influence of *cis*-acting SVA RIPs on gene regulation, we next analysed the gene expression patterns of the top two hits for the largest effects on gene upregulation and downregulation (Appendix A). The presence of either one (PA) or two (PP) SVA_67 alleles significantly upregulated both *LRRC37A4P* and *MAPT* gene expressions when tested using the Wilcoxon pairwise comparison with FDR-adjusted p values (FDR *p* < 0.05) (Appendix A). In comparison to individuals homozygous absent for SVA_67, homozygous present and heterozygous individuals displayed a 447.3-fold (*p* = 1.04 × 10^−40^) and a 214.4-fold (*p* = 2.55 × 10^−38^) increase in *LRRC37A4P* expressions and a 1.4-fold (*p* = 2.53 × 10^−9^) and 1.2-fold (*p* = 2.51 × 10^−4^) increase in *MAPT* gene expressions, respectively (Appendix A). An analysis of the top two hits for the greatest effect on gene downregulation displayed differing gene expression patterns between the influence of SVA_73 and SVA_72 on *FCGBP* gene expressions (Appendix A). When comparing to the complete absence (AA) of SVA_73, the PA and PP genotype groups displayed a significant 3.7-fold (*p* = 3.61 × 10^−2^) and 10-fold (*p* = 2.61 × 10^−3^) increase in *FCGBP* gene expressions (Appendix A). Although our analysis displayed that the presence of SVA_73 upregulates the *FCGBP* gene in comparison to AA individuals, *FCGBP* gene expressions are significantly repressed by 63% (*p* = 3.29 × 10^−8^) in the PP genotype group compared to the PA genotype group (Appendix A). An analysis of SVA_72 highlighted that the homozygous presence of the SVA_72 allele significantly downregulated *FCGBP* gene expressions by 23% (*p* = 5.74 × 10^−3^) in comparison to its homozygous absence (Appendix A). PP genotype individuals also displayed a significant reduction of 32% (*p* = 1.85 × 10^−5^) in *FCGBP* gene expressions in comparison to the PA genotype (Appendix A). Similar to the SVA_73 gene expression analysis, individuals heterozygous for SVA_72 indicated a 1.2-fold (*p* = 4.23 × 10^−1^) increase in *FCGBP* expressions; however, statistical significance was not achieved (Appendix A).

As a recent genome-wide association study (GWAS) reported an association of the *HLA* locus with ALS risk [43,44], we investigated the influence that SVA RIPs have on *cis*-regulated *HLA* genes (Appendix A). This analysis demonstrated that the top ten hits for the greatest effect of SVA RIPs regarding upregulation was for *HLA* genes. We discovered a common gene expression pattern across all *HLA* genes analysed, demonstrating that individuals homozygous present for SVA RIPs (SVA_24, SVA_25, SVA_27, and SVA_88) significantly upregulate *HLA* genes in comparison to homozygous-absent individuals (*p* < 0.01) (Appendix A).

### 3.4. SVA RIPs Display Tissue-Specific Modulations of Gene Expressions

To investigate tissue-specific influences of reference SVA polymorphisms, we conducted an eQTL analysis on the CNS tissue types of spinal cord, cerebellum, motor cortex, and frontal cortex individually (Table 2 and Table 3). The temporal cortex, occipital cortex, and hippocampus tissues were excluded from this analysis due to the low *n* numbers. The result tables display the eQTL analysis from each individual tissue to allow for the determination of the top 40 hits for greatest effect size on gene upregulation (positive beta values) (Table 2) and downregulation (negative beta values) (Table 3). From this analysis, we established that a large proportion of the gene regulation effects were present within spinal cord tissues, displaying 27/40 and 26/40 tissue-specific hits for both gene upregulation and downregulation, respectively (Table 2 and Table 3). Furthermore, our analysis revealed that five SVAs (SVA_55, SVA_15, SVA_37, SVA_85, and SVA_4) were responsible for the 40 most positive beta values, and eight SVAs (SVA_90, SVA_5, SVA_87, SVA_93, SVA_30, SVA_70, SVA_84, SVA_91, SVA_16) were responsible for the 40 most negative beta values, all effects of which were in the *trans* position (Table 2 and Table 3). These data demonstrate the capability of SVAs to modulate the expression of multiple gene targets.

Moreover, the *MBP* gene located on chromosome 18 and the *PLP1* gene located on the X chromosome were the most significantly upregulated and downregulated genes simultaneously (Table 2 and Table 3). Here, the gene expression of both genes was modulated in spinal cord tissues (Table 2 and Table 3). In addition, the genes *MTURN* and *MOBP* and mitochondrial genes (*MTCO1P12*, *MT-ND1*, *MT-ND2*, *MT-ND3*, *MT-ND4*, *MT-ND5*, *MT-CO2*, *MT-CO3*, and *MT-CYB*) appear multiple times within both gene upregulation and downregulation hits (Table 2 and Table 3). Upon an analysis of the whole tissue-specific dataset, the genes *MBP*, *PLP1*, *MTURN*, and *MOBP* were only regulated within spinal cord tissues, whilst the expression of mitochondrial genes (encoded by mitochondrial DNA) was modulated within spinal cord, cerebellum, and motor cortex tissues, while *MTCO1P12* gene expressions were regulated in all four tissue types (Table 2 and Table 3). Thus, this illustrates that SVAs possess the capacity to influence tissue-specific gene expressions.

## 4. Discussion

In this study, we evaluated the role of reference SVAs polymorphic for their presence in the human genome to modulate gene expressions within CNS tissues of ALS patients and healthy controls. An analysis of the NYGC ALS Consortium dataset demonstrated that SVA RIPs significantly regulate gene expressions genome-wide and in a tissue-specific manner. This study continues to expand on our previous findings, demonstrating the capability of SVAs to differentially regulate gene expressions [31,42]. We have previously illustrated the capacity of SVAs to influence gene expressions within Parkinson’s disease, highlighting the role of SVA_67 to modulate genes at the *MAPT* locus within the PPMI cohort and a CRISPR deletion model [27,32,40]. This analysis further validates our previous research, illustrating the functional capacity of SVA_67 within neurodegenerative diseases and potentially expanding the importance of SVA_67 and the *MAPT* locus to ALS (Figure 5 and Figure 6). Therefore, this study not only emphasizes the correlation between SVA presence or absence and differential gene expressions but also the involvement of SVAs within disease pathology.

Using WGS data from the ALS Consortium for a matrix eQTL analysis, we not only identified that polymorphic SVA RIPs possess the ability to modify the expression of multiple target genes, but also many SVAs can regulate the gene expression of a single target. Of these SVA RIPs, a greater proportion of *trans*-regulatory effects were displayed in comparison to *cis* ones. Similarly, previous eQTL studies analysing RNA-seq data from the 1000 genome project and the PPMI cohort, respectively, identified the impact of TEs on gene expressions [42,45]. In line with our analysis, these studies highlighted that several TEs simultaneously modulate the gene expression of a single gene, that an individual TE can regulate the expression of multiple genes, and that a greater number of the TEs analysed were in the *trans* position [42,45]. A potential mechanism for *trans*-acting eQTLs could be through the binding of the CCCTC-binding factor (CTCF) to SVAs. It has been demonstrated that CTCF can bind the SVA VNTR in vitro, while the germline-expressed paralog CTCF-like can do this in vivo [46]. Through the co-operation with protein complex cohesion, CTCF plays a key role in three-dimensional chromatin regulation and chromatin looping, bringing promoters and regulatory elements within close proximity to activate or repress gene expressions [46,47]. An additional mechanism could be through indirect transcription factor (TF)-mediated associations. This suggests that SVAs influencing the expression of TF could indirectly regulate one or multiple TF gene targets, ultimately activating or repressing TF target gene expressions [45].

Upon the examination of beta values from the eQTL analysis, we determined that within the combined analysis of CNS tissues, SVA_55 demonstrated the greatest effect on myelin basic protein (*MBP*) gene upregulation (Figure 4). Following further analysis, we identified that through individual tissue analyses, the top hits for gene upregulation and downregulation were SVAs influencing *MBP* gene expressions (Table 2 and Table 3). However, *MBP* expressions were only significantly modulated within spinal cord tissues. MBP is a key protein involved in the myelination process, whereby myelin sheaths are formed around CNS axons by oligodendrocytes [48,49]. As oligodendrocyte loss and myelin dysfunction has recently been emphasized in neurodegenerative diseases, including ALS, it is essential to investigate the relationship between SVAs and *MBP* differential expressions [50,51]. Lorente Pons et al. demonstrated the potential significance of MBP in ALS through a post-mortem analysis of both sporadic ALS and C9orf72-related ALS cases, identifying a significant reduction in MBP protein abundance when normalised to the proteolipid protein (PLP) in the spinal cord corticospinal tracts in ALS cases in comparison to controls [52]. As *MBP* mRNA is transported to the myelin compartment by the RNA transport granule and PLP is transported as a protein, this suggests that the reduction in MBP could be due to an impaired mRNA transport [52]. Our data suggests that certain SVAs (SVA_90 and SVA_5) act to downregulate *MBP* gene expressions; therefore, SVAs could play a role in this mechanism, leading to the reduction in MBP protein levels observed in the ALS patients compared to the controls.

Our analysis also demonstrated that SVAs can activate (SVA_37) and repress (SVA_87 and SVA_93) *PLP1*, a form of PLP. *PLP1* has previously been implicated in Pelizaeus–Merzbacher disease (PMD), an X-linked neurodegenerative disease, whereby mutations within this gene inhibit CNS myelination [53,54]. In addition, we demonstrate that the myelin-associated oligodendrocyte basic protein (*MOBP*) gene, the locus of which has been highlighted for ALS risk, is again potentially regulated by multiple SVAs (SVA_5, 15, 37, 55, 84, 85, 87, 91, and 93) only within the spinal cord [43]. Hence, SVA regulation could be a potential mechanism involved in ALS risk at the *MOBP* locus. Furthermore, our tissue-specific analysis showed that multiple mitochondrial genes (*MT-ND1,2,3,4,5, MT-CYB*, *MT-CO2,3*, and pseudogene *MTCO1P12*) were largely activated and repressed within all four tissues analysed. SVA_30 and SVA_70 displayed the greatest effects on mitochondrial genes, with SVA_30 modulating five targets and SVA_70 affecting four gene targets. As mitochondrial dysfunction is known to be implicated in other neurodegenerative diseases, including PD and Alzheimer’s disease as well as ALS, SVA regulation resulting in differential expressions could be an underlying mechanism involved in disease pathology [55,56].

Upon an analysis of CNS tissues from both ALS individuals and healthy controls, seven of the top ten *cis*-acting reference SVA RIPs imposed the greatest effects on the upregulation of genes at the human leukocyte antigen (*HLA*) locus (Figure 6a). Four SVAs are responsible for the effects on *HLA* gene expressions, whereby SVA_24 influences the expression of one gene (*HLA-A*), SVA_25 two genes (*HLA-B* and *HLA-C*), SVA_27 two genes (*HLA-DRB1* and *HLA-DRB5*), and SVA_88 one gene (*HLA-DQB1*). HLA, also referred to as the major histocompatibility complex (MHC), acts to regulate both innate and adaptive immunity involved in the human immune response system [57]. The involvement of the immune response system in neurological diseases has been recognised, and the *HLA* locus has been highlighted as a region of importance in numerous neurodegenerative diseases, including ALS [57]. Various studies have investigated the significance and mechanism of HLA in ALS, demonstrating increased frequencies of *HLA-A*, *HLA-B*, and *HLA-C* alleles in ALS cases compared to controls [58,59,60,61]. A recent large-scale GWAS conducted by Van Rheenen et al. identified the *HLA* region as a locus significantly associated with ALS, further highlighting the importance of this locus [43]. Previous studies within our group have highlighted the capability of SVAs to modulate *HLA* gene expressions; analyses of whole-genome sequencing and transcriptomic data obtained from the whole blood of individuals within a PPMI cohort discovered that SVA_24, SVA_25, and SVA_27 modulate the expression of *HLA-A*, *HLA-B*, and *HLA-C* and *HLA-DRB1* and *HLA-DRB5*, respectively [42,62]. This suggests that the modulation of *HLA* genes by SVAs could be a common mechanism within neurodegenerative diseases.

In conclusion, we show that SVAs demonstrate a significant impact on the expression of individual or numerous genes, including those previously associated with neurodegenerative diseases, such as ALS. Ultimately, the ability of SVAs to act as a regulatory domain can be an underappreciated source of genetic variation, and this may be involved in the mechanisms leading to diseases and disease progression. Here, differential gene expressions caused by TE polymorphism could be one mechanism. However, due to limitations, such as low *n* numbers for some SVA genotypes and CNS tissue types (occipital cortex, temporal cortex, and hippocampus), further research investigating the involvement of TEs in the pathogenesis of neurodegenerative diseases, specifically ALS, is crucial. In addition, due to the low *n* numbers in the ALS and control groups, an individual group analysis was not possible. Therefore, future experiments individually investigating the influence of SVAs in ALS patient and control groups is required. Furthermore, although this analysis continues to demonstrate the potential role of SVAs in neurodegenerative diseases, further experiments, such as those involving CRISPR, are essential to validate SVA-specific influences. For example, we have previously shown that SVA_67 deletion in a CRISPR model resulted in a significant increase in *MAPT* and *LRRC37A* gene expressions [40].

## Figures and Tables

**Figure 1 biomolecules-14-00358-f001:**
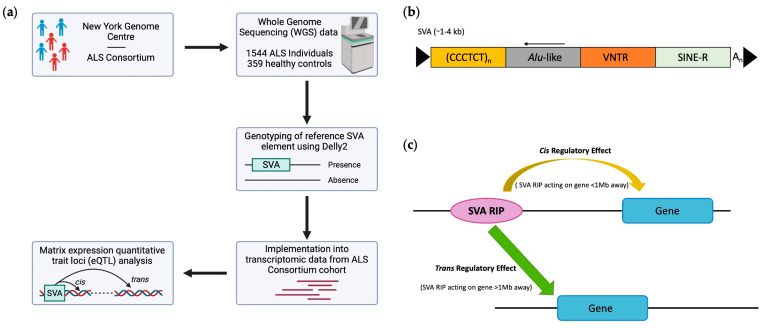
General overview of this study, including the structure of a full-length SVA element and both *cis*- and *trans*-acting mechanisms. (**a**) This study incorporated whole-genome sequencing and transcriptomic data from the New York Genome Centre ALS Consortium cohort to investigate the ability of SVA retrotransposon insertion polymorphisms (RIPs) to act as expression quantitative trait loci (eQTL) within central nervous system (CNS) tissues. (**b**) Schematic of SVA structure, displaying a full-length SVA element consisting of a 5′ CT-rich hexamer repeat, an *Alu*-like region, a variable-number tandem repeat (VNTR), a SINE (short interspersed nuclear element)-R domain, and a 3′ poly-A tail. (**c**) The mechanism by which SVAs implement a *cis*- or *trans*-regulatory effect. *Cis*-regulatory effects are defined as effects observed by elements (SVA RIPs) which act to modulate the expression of genes less than 1 Mb away from the element site, whilst *trans*-regulatory effects are defined as effects observed by elements which act to modulate the expression of genes greater than 1 Mb away from the element site.

**Figure 2 biomolecules-14-00358-f002:**
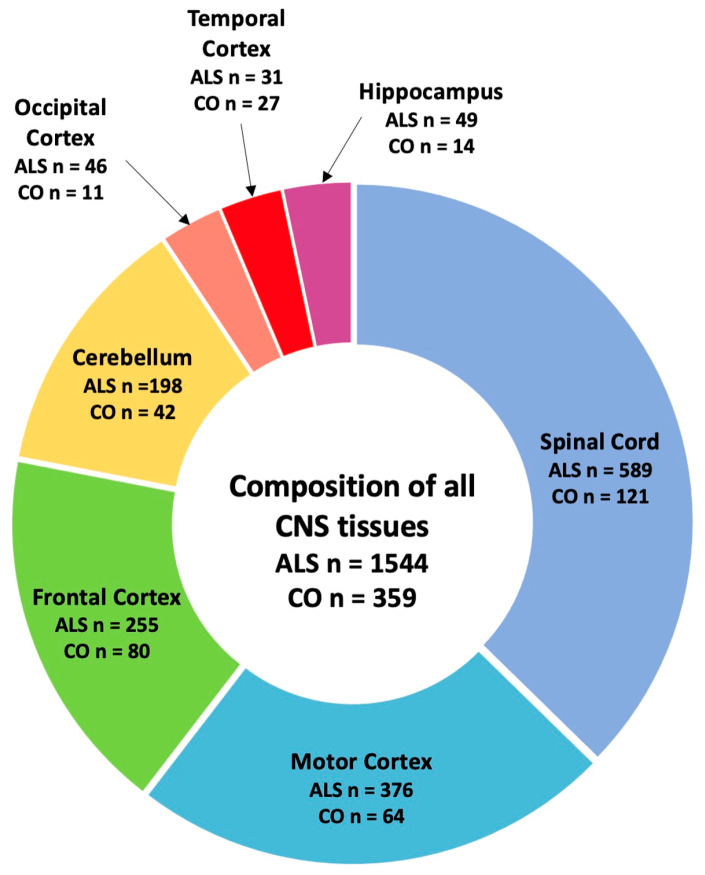
Composition of tissues used in this study. For this study, CNS tissue data from healthy controls (COs) and ALS patients combined (*n* = 1903), which composed of spinal cord (*n* = 710), motor cortex (*n* = 440), frontal cortex (*n* = 335), cerebellum (*n* = 240), occipital cortex (*n* = 57), temporal cortex (*n* = 58), and hippocampus (*n* = 63), were included for our analysis.

**Figure 3 biomolecules-14-00358-f003:**
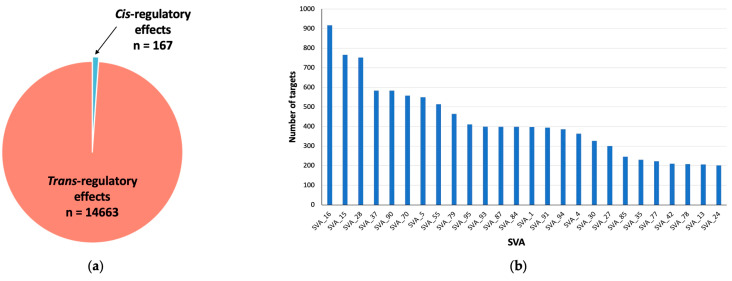
Overview of the number of genomic loci affected by SVA polymorphism following a matrix eQTL analysis of all CNS tissues. (**a**) Pie chart representing the composition of all significant differentially regulated genetic loci (*n* = 14,830), displaying the number of *cis*-regulatory (*n* = 167) and *trans*-regulatory (*n* = 14,663) effects exhibited by SVAs within all CNS tissues. (**b**) Bar chart displaying reference SVAs and the number of genome-wide gene targets. Each of the 92 analysed SVAs had a significant impact on multiple targets, with the lowest number of targets for one SVA being 6 (FDR *p* < 0.05). Only SVAs affecting more than 200 targets are displayed (*n* = 26). For this analysis, data from ALS individuals and healthy controls were combined.

**Figure 4 biomolecules-14-00358-f004:**
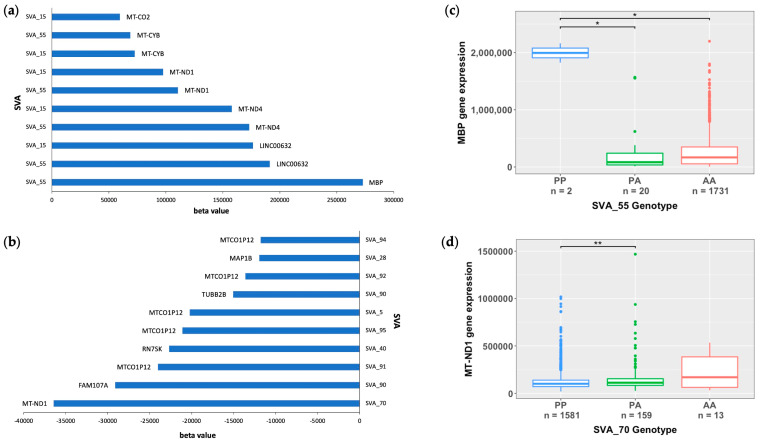
Reference SVA RIP elements with the greatest effect size from matrix eQTL analysis of all CNS tissues. (**a**,**b**) Clustered bar chart showing the top ten reference SVA RIPs across all CNS tissues with the greatest effect size on gene upregulation (positive beta values) (**a**) and gene downregulation (negative beta values) (**b**) from an eQTL analysis. SVA_55 demonstrated the greatest increase in the activation of the *MBP* gene with a beta coefficient of 272,847, whilst SVA_70 demonstrated the greatest repressive effect on the *MT-ND1* gene with a beta coefficient of −36,441. (**c**) Boxplot of SVA_55 indicating *MBP* gene expressions stratified by the SVA_55 genotype. Genotypes PP (*n* = 2), PA (*n* = 20) and AA (*n* = 1731). Significant differences in *MBP* gene expressions were observed between the PP and PA group (*p* = 0.0218) and the PP and AA group (*p* = 0.0218). Subjects with the PP genotype displayed a 9.5-fold and 19.2-fold increase in *MBP* gene expressions in comparison to subjects with AA and PA genotypes, respectively. (**d**) Boxplot of SVA_70 displaying *MT-ND1* gene expressions, stratified by the SVA_70 genotype. Genotypes PP (*n* = 1581), PA (*n* = 159), and AA (*n* = 13). A significant repression in *MT-ND1* gene expressions of 55% was observed between the PP and PA subject groups (*p* = 0.0098). No statistical significance was obtained for differences between the PP and PA subject groups. For both boxplots, the significance of gene expression changes between groups was determined using the Wilcoxon pairwise comparison with FDR-adjusted *p* values (FDR *p* < 0.05). The PP, PA, and AA groups represent when there are two copies of the SVA present, one copy of the SVA present, and the complete absence of the SVA, respectively. * *p* < 0.05, ** *p* < 0.01.

**Figure 5 biomolecules-14-00358-f005:**
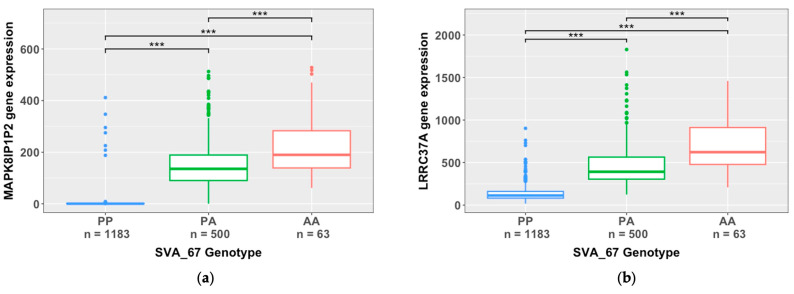
Boxplots of two of the most significant SVA_67 interactions with *MAPK8IP1P2* and *LRRC37A* obtained from a matrix eQTL analysis. Both significant effects are *cis*-regulatory effects. Datapoints from both ALS individuals and healthy controls across all CNS tissues were combined, and the significance of gene expression changes between groups was determined using the Wilcoxon pairwise comparison with FDR-adjusted p values (FDR < 0.05). (**a**) Boxplot showing the association of the SVA_67 genotype with *MAPK8IP1P2* gene expressions. Significant differences were observed between all groups for *MAPK8IP1P2* expressions, namely PP and PA (*p* = 7.33 × 10^−237^), PP and AA (*p* = 4.26 × 10^−55^), and PA and AA (*p* = 1.33 × 10^−7^). (**b**) Boxplot showing the association of the SVA_67 genotype with *LRRC37A* gene expressions. Significant differences were observed between all groups for *LRRC37A* gene expressions, namely PP and PA (*p* = 1.95 × 10^−204^), PP and AA (*p* = 1.06 × 10^−39^), and PA and AA (*p* = 2.13 × 10^−12^). Fold change in expressions of 262-fold and 6-fold for *MAPK8IP1P2* and *LRRC37A*, respectively, was observed for individuals with AA genotypes in comparison to PP genotypes. PP (*n* = 1183), PA (*n* = 500), and AA (*n* = 63). *** *p* < 0.001.

**Figure 6 biomolecules-14-00358-f006:**
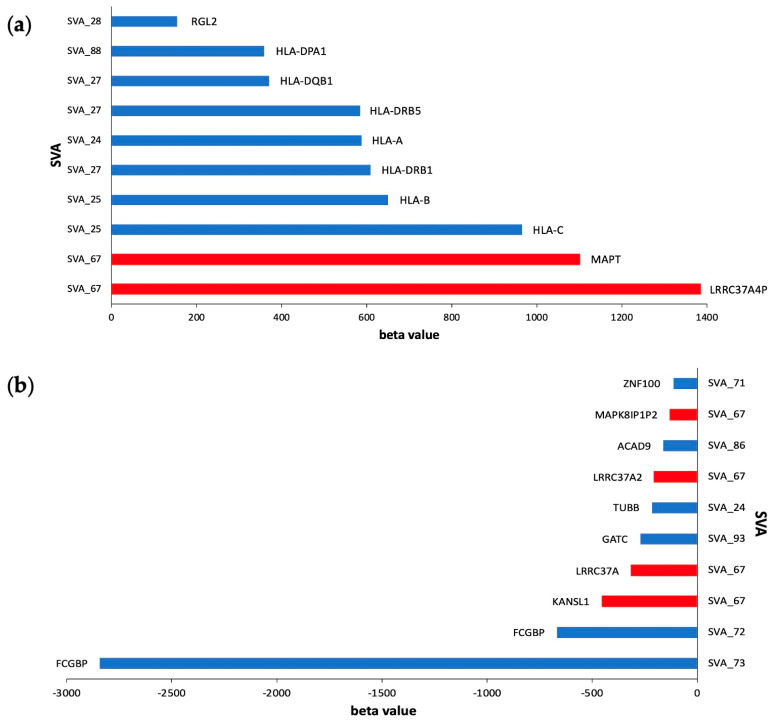
Top ten *cis*-acting reference SVA elements with the greatest effects on gene upregulation and downregulation in CNS tissues. (**a**) Clustered bar chart displaying ten reference SVA RIPs with the greatest *cis*-regulatory effects on gene activation (most positive beta values) from a matrix eQTL analysis (FDR *p* < 0.05). SVA_67 is responsible for the two greatest increases in the activation of the genes *LRRC37A4P* and *MAPT*, displaying beta coefficients of 1384 and 1101, respectively. SVA_24 (*HLA-A*); SVA_25 (*HLA-C* and *HLA-B*); SVA_27 (*HLA-DRB1*, *HLA-DRB5*, and *HLA-DQB1*); and SVA_88 (*HLA-DPA1*) were shown to be responsible for the large increases in the activation of a series of *HLA* genes. (**b**) Clustered bar chart displaying ten reference SVA RIPs with the greatest *cis*-regulatory effects on gene downregulation (most negative beta values) from a matrix eQTL analysis (FDR *p* < 0.05). The two greatest repressive effects were on the *FCGBP* gene, regulated by both SVA_73 and SVA_72, demonstrating a beta coefficient of −2843 and −668, respectively. SVA_67 was responsible for four of these effects, by showing a downregulating effect, for the genes *KANSL1*, *LRRC37A*, *LRRC37A2*, and *MAPK8IP1P2*. This analysis combined datapoints from ALS individuals and healthy controls.

**Table 1 biomolecules-14-00358-t001:** Top ten most significant reference SVA RIP effects from a matrix eQTL analysis of all CNS tissues. This analysis included the combination of both *cis* and *trans* effects as well as datapoints from both ALS individuals and healthy controls.

SVA	Beta Value	False Discovery Rate (FDR)	Target Gene	*Cis/trans* Effect
SVA_67	−131.2	1.93 × 10^−303^	*MAPK8IP1P2*	*cis*
SVA_67	−5.2	1.93 × 10^−303^	*ENSG00000285668.1*	*cis*
SVA_67	−315.1	5.12 × 10^−299^	*LRRC37A*	*cis*
SVA_87	−126.2	3.02 × 10^−224^	*MTND4P24*	*trans*
SVA_93	−126.2	4.22 × 10^−224^	*MTND4P24*	*trans*
SVA_84	−63.1	4.22 × 10^−224^	*MTND4P24*	*trans*
SVA_58	4.1	3.89 × 10^−211^	*LLPH-DT*	*cis*
SVA_24	12.3	5.06 × 10^−201^	*HLA-K*	*cis*
SVA_15	59.6	3.13 × 10^−189^	*MTND4P24*	*trans*
SVA_33	48.4	8.84 × 10^−188^	*ZFAND2A-DT*	*cis*

**Table 2 biomolecules-14-00358-t002:** Top 40 significant reference SVA RIPs with the greatest effects on gene upregulation (most positive beta values) from a tissue-specific matrix eQTL analysis. This analysis included the combination of both *cis* and *trans* effects as well as datapoints from both ALS individuals and healthy controls.

SVA	Gene ID	FDR *p* Value	Beta Value	Gene	Chr	*Cis/trans*	Tissue
SVA_55	ENSG00000197971.16	5.99 × 10^−10^	638,804.662	*MBP*	18	*trans*	Spinal Cord
SVA_15	ENSG00000197971.16	3.59 × 10^−7^	620,026.724	*MBP*	18	*trans*	Spinal Cord
SVA_37	ENSG00000197971.16	8.76 × 10^−4^	585,791.502	*MBP*	18	*trans*	Spinal Cord
SVA_85	ENSG00000197971.16	2.08 × 10^−3^	560,691.911	*MBP*	18	*trans*	Spinal Cord
SVA_37	ENSG00000123560.14	3.72 × 10^−12^	168,441.474	*PLP1*	X	*trans*	Spinal Cord
SVA_55	ENSG00000198888.2	1.47 × 10^−5^	153,947.476	*MT-ND1*	MT	*trans*	Spinal Cord
SVA_55	ENSG00000203930.12	5.97 × 10^−4^	148,616.177	*LINC00632*	X	*trans*	Motor Cortex
SVA_15	ENSG00000198888.2	1.62 × 10^−3^	142,961.051	*MT-ND1*	MT	*trans*	Spinal Cord
SVA_55	ENSG00000203930.12	1.25 × 10^−16^	115,977.491	*LINC00632*	X	*trans*	Spinal Cord
SVA_15	ENSG00000203930.12	1.86 × 10^−8^	969,10.7666	*LINC00632*	X	*trans*	Spinal Cord
SVA_55	ENSG00000180354.16	1.34× 10^−22^	90,266.4311	*MTURN*	7	*trans*	Spinal Cord
SVA_15	ENSG00000123560.14	3.43 × 10^−3^	80,380.5623	*PLP1*	X	*trans*	Spinal Cord
SVA_85	ENSG00000203930.12	4.76 × 10^−3^	78,457.5282	*LINC00632*	X	*trans*	Spinal Cord
SVA_15	ENSG00000180354.16	1.86 × 10^−12^	77,839.9088	*MTURN*	7	*trans*	Spinal Cord
SVA_55	ENSG00000198712.1	3.04 × 10^−3^	76,318.8326	*MT-CO2*	MT	*trans*	Spinal Cord
SVA_85	ENSG00000180354.16	1.35 × 10^−5^	67,900.9382	*MTURN*	7	*trans*	Spinal Cord
SVA_85	ENSG00000237973.1	4.10 × 10^−33^	66,702.2584	*MTCO1P12*	1	*trans*	Motor Cortex
SVA_37	ENSG00000168314.18	3.50 × 10^−9^	65,555.4973	*MOBP*	3	*trans*	Spinal Cord
SVA_85	ENSG00000237973.1	1.01 × 10^−32^	53,997.5748	*MTCO1P12*	1	*trans*	Frontal Cortex
SVA_15	ENSG00000168314.18	5.02 × 10^−9^	53,362.8451	*MOBP*	3	*trans*	Spinal Cord
SVA_55	ENSG00000237973.1	3.61 × 10^−22^	52,048.0183	*MTCO1P12*	1	*trans*	Motor Cortex
SVA_55	ENSG00000168314.18	1.38 × 10^−9^	49,216.1425	*MOBP*	3	*trans*	Spinal Cord
SVA_15	ENSG00000237973.1	2.83 × 10^−26^	48,449.3666	*MTCO1P12*	1	*trans*	Motor Cortex
SVA_85	ENSG00000168314.18	3.76 × 10^−4^	47,350.2242	*MOBP*	3	*trans*	Spinal Cord
SVA_85	ENSG00000237973.1	8.82 × 10^−42^	39,521.0532	*MTCO1P12*	1	*trans*	Spinal Cord
SVA_55	ENSG00000064787.13	1.62 × 10^−30^	36,504.4209	*BCAS1*	20	*trans*	Spinal Cord
SVA_37	ENSG00000091513.16	1.65× 10^−6^	35,838.5557	*TF*	3	*trans*	Spinal Cord
SVA_55	ENSG00000237973.1	1.49 × 10^−59^	33,728.2747	*MTCO1P12*	1	*trans*	Spinal Cord
SVA_4	ENSG00000237973.1	3.59 × 10^−17^	32,837.5825	*MTCO1P12*	1	*trans*	Frontal Cortex
SVA_37	ENSG00000237973.1	1.52 × 10^−23^	32,783.3654	*MTCO1P12*	1	*trans*	Motor Cortex
SVA_37	ENSG00000237973.1	1.72 × 10^−31^	32,728.512	*MTCO1P12*	1	*trans*	Frontal Cortex
SVA_15	ENSG00000237973.1	8.40 × 10^−41^	32,193.3558	*MTCO1P12*	1	*trans*	Spinal Cord
SVA_37	ENSG00000099194.6	2.82 × 10^−4^	31,333.7253	*SCD*	10	*trans*	Spinal Cord
SVA_85	ENSG00000064787.13	3.78 × 10^−10^	30,888.7058	*BCAS1*	20	*trans*	Spinal Cord
SVA_15	ENSG00000237973.1	3.46 × 10^−14^	30,602.307	*MTCO1P12*	1	*trans*	Frontal Cortex
SVA_15	ENSG00000064787.13	1.97 × 10^−15^	30,356.5865	*BCAS1*	20	*trans*	Spinal Cord
SVA_37	ENSG00000237973.1	1.28 × 10^−6^	30,025.949	*MTCO1P12*	1	*trans*	Cerebellum
SVA_15	ENSG00000237973.1	5.29 × 10^−13^	29,472.617	*MTCO1P12*	1	*trans*	Cerebellum
SVA_55	ENSG00000237973.1	2.00 × 10^−8^	27,693.2393	*MTCO1P12*	1	*trans*	Frontal Cortex
SVA_4	ENSG00000237973.1	2.56 × 10^−9^	27,128.1241	*MTCO1P12*	1	*trans*	Motor Cortex

**Table 3 biomolecules-14-00358-t003:** Top 40 significant reference SVA RIPs with the greatest effects on gene downregulation (most negative beta values) from a tissue-specific matrix eQTL analysis. This analysis included the combination of both *cis* and *trans* effects as well as datapoints from both ALS individuals and healthy controls.

SVA	Gene ID	FDR *p*-Value	Beta Value	Gene	Chr	*Cis/trans*	Tissue
SVA_90	ENSG00000197971.16	3.28 × 10^−3^	−1,337,611.1	*MBP*	18	*trans*	Spinal Cord
SVA_5	ENSG00000197971.16	3.06 × 10^−3^	−351,064.11	*MBP*	18	*trans*	Spinal Cord
SVA_87	ENSG00000123560.14	1.98 × 10^−7^	−193,423.77	*PLP1*	X	*trans*	Spinal Cord
SVA_93	ENSG00000123560.14	2.05 × 10^−7^	−193,266.83	*PLP1*	X	*trans*	Spinal Cord
SVA_30	ENSG00000198888.2	8.47 × 10^−3^	−143,159.48	*MT-ND1*	MT	*trans*	Cerebellum
SVA_70	ENSG00000198886.2	1.029 × 10^−2^	−133,464.74	*MT-ND4*	MT	*trans*	Cerebellum
SVA_30	ENSG00000198763.3	7.85 × 10^−4^	−131,182.45	*MT-ND2*	MT	*trans*	Motor Cortex
SVA_30	ENSG00000198938.2	6.86 × 10^−3^	−111,339.82	*MT-CO3*	MT	*trans*	Cerebellum
SVA_30	ENSG00000198727.2	5.66 × 10^−3^	−104,147.35	*MT-CYB*	MT	*trans*	Cerebellum
SVA_84	ENSG00000123560.14	2.11 × 10^−7^	−96,556.036	*PLP1*	X	*trans*	Spinal Cord
SVA_30	ENSG00000198786.2	7.25 × 10^−3^	−94,294.133	*MT-ND5*	MT	*trans*	Motor Cortex
SVA_70	ENSG00000198763.3	1.29 × 10^−2^	−82,929.066	*MT-ND2*	MT	*trans*	Cerebellum
SVA_90	ENSG00000259001.3	6.14 × 10^−3^	−73,740.939	*ENSG00000259001*	14	*trans*	Spinal Cord
SVA_91	ENSG00000203930.12	2.02 × 10^−3^	−71,147.071	*LINC00632*	X	*trans*	Spinal Cord
SVA_87	ENSG00000168314.18	3.60 × 10^−4^	−67,162.942	*MOBP*	3	*trans*	Spinal Cord
SVA_93	ENSG00000168314.18	3.67 × 10^−4^	−67,112.229	*MOBP*	3	*trans*	Spinal Cord
SVA_91	ENSG00000180354.16	2.99 × 10^−5^	−57,438.97	*MTURN*	7	*trans*	Spinal Cord
SVA_90	ENSG00000168309.18	4.26 × 10^−3^	−48,145.904	*FAM107A*	3	*trans*	Spinal Cord
SVA_5	ENSG00000180354.16	1.55 × 10^−5^	−43,141.656	*MTURN*	7	*trans*	Spinal Cord
SVA_70	ENSG00000198712.1	6.03 × 10^−3^	−43,049.326	*MT-CO2*	MT	*trans*	Cerebellum
SVA_90	ENSG00000168309.18	2.45 × 10^−3^	−40,637.548	*FAM107A*	3	*trans*	Motor Cortex
SVA_91	ENSG00000168314.18	5.48 × 10^−4^	−40,457.332	*MOBP*	3	*trans*	Spinal Cord
SVA_90	ENSG00000177575.13	2.99 × 10^−19^	−36,552.585	*CD163*	12	*trans*	Spinal Cord
SVA_84	ENSG00000168314.18	3.83 × 10^−4^	−33,494.891	*MOBP*	3	*trans*	Spinal Cord
SVA_90	ENSG00000087086.15	1.64 × 10^−8^	−32,820.222	*FTL*	19	*trans*	Motor Cortex
SVA_5	ENSG00000168314.18	1.72 × 10^−4^	−31,165.253	*MOBP*	3	*trans*	Spinal Cord
SVA_16	ENSG00000123560.14	8.97 × 10^−7^	−31,144.963	*PLP1*	X	*trans*	Spinal Cord
SVA_91	ENSG00000237973.1	6.16 × 10^−30^	−29,834.037	*MTCO1P12*	1	*trans*	Spinal Cord
SVA_87	ENSG00000173786.17	5.88 × 10^−4^	−28,556.333	*CNP*	17	*trans*	Spinal Cord
SVA_93	ENSG00000173786.17	5.98 × 10^−4^	−28,536.883	*CNP*	17	*trans*	Spinal Cord
SVA_90	ENSG00000137285.11	1.48 × 10^−40^	−27,173.64	*TUBB2B*	6	*trans*	Motor Cortex
SVA_5	ENSG00000237973.1	3.92 × 10^−10^	−24,995.474	*MTCO1P12*	1	*trans*	Motor Cortex
SVA_5	ENSG00000237973.1	8.77 × 10^−13^	−24,871.705	*MTCO1P12*	1	*trans*	Frontal Cortex
SVA_91	ENSG00000064787.13	5.47 × 10^−7^	−22,960.148	*BCAS1*	20	*trans*	Spinal Cord
SVA_93	ENSG00000198840.2	3.12 × 10^−3^	−22,049.981	*MT-ND3*	MT	*trans*	Spinal Cord
SVA_87	ENSG00000198840.2	3.13 × 10^−3^	−22,046.442	*MT-ND3*	MT	*trans*	Spinal Cord
SVA_90	ENSG00000164733.22	1.39 × 10^−4^	−21,828.547	*CTSB*	8	*trans*	Spinal Cord
SVA_90	ENSG00000079215.15	1.69 × 10^−5^	−21,020.491	*SLC1A3*	5	*trans*	Motor Cortex
SVA_5	ENSG00000237973.1	8.40 × 10^−24^	−19,907.219	*MTCO1P12*	1	*trans*	Spinal Cord
SVA_87	ENSG00000136541.15	1.45 × 10^−10^	−19,551.323	*ERMN*	2	*trans*	Spinal Cord

## Data Availability

The sequencing (RNA and WGS) data analysed in this study from the ALS Consortium were obtained upon application to the New York Genome Centre, and data requests can be made by completing a genetic data request form at ALSData@nygenome.org. Additional data from this study will be made available upon reasonable request.

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
