# Peer review of "Exploring SVA Insertion Polymorphisms in Shaping Differential Gene Expressions in the Central Nervous System"

_biomolecules, 2024, doi:10.3390/biom14030358_

Round 1
Reviewer 1 Report
Comments and Suggestions for Authors
Hughes et al. study how SVA elements regulate gene expression in the central nervous system. They use a cohort of approximately 1500 ALS cases and 360 controls. By using both whole-genome sequencing and transcriptomics data they identify SVAs that regulate thousands of genes. The article is well-written and elucidates an understudied mechanism of gene expression regulation. The authors have previously responded to several review comments so I don’t have too many comments:
Major comments
1. The authors use data that they did not produce from the New York Genome Center. Therefore, it is important to assess that in this particular dataset that their n is sufficient to control potential biases as there might be some patient recruitment or technical details that the NYGC does not clearly communicate that result in unexpected subgroups. I suggest that the authors:
a. perform e.g. a hierarchical clustering analysis of the transcriptomic data to check if there are any “hidden” subgroups in the data that need to be taken into account
b. use the WGS data to create principal components to control for population stratification and rerun their analyses with these PCs as covariates (in addition to age and sex they have already used) and see if the results remain largely the same
c. The authors can then add their observations from the above points in the limitations section in Discussion if necessary
2. There is no replication data. This might not be feasible as SVAs are not routinely genotyped and not all SVAs are reliably tagged by SNPs so replicating their results can be difficult. However, if the authors can identify SNPs that are in LD with a couple of their most interesting SVAs and replicate their eQTL associations with e.g. GTEx portal data, that would really strengthen their results.
Minor comments:
1. Building on previous review comments and responses, it would be good to include the fact they have benchmarked Delly’s performance on their SVAs to the method’s section to highlight the reliability of their SVA genotypes
2. In one of the earlier reviewer comment responses the authors show that the results of this work are also supported by their previous SVA work. I think they could highlight this additional layer of evidence a bit more in the Discussion.
Author Response
We thank all reviewers for the comments that helped us to improve and clarify our statements and results in the manuscript. Our responses are the following:
- To answer these concerns, we have to understand that we are using real-world patient-derived data here. These are post-mortem data, and the stratification, or cell population distribution, if existing, is impossible to control in any realistic way. However, we can assure the reviewer that our sample size is so large (1544 patients and 359 controls) that any systemic noise, like differences in the cell population distributions, will be balanced. This is the main point of using large sample sizes: they resist random noise or systemic biases. We have done a principal component analysis (figure attached) of the RNA-seq data, and the tissue was the main component that affected the whole transcriptome. This is the spinal cord or cerebellum frontal cortex, etc. The tissue effect was even stronger than the effect of sex on the whole transcriptome. We did not find any evidence for the population stratification in the whole transcriptome data.

2. We have the replication data and have already published an independent replication analysis. We have done similar analyses on an entirely independent cohort of PPMI and have published many papers based on these findings. We have also indicated a causal link between the SVAs and gene expression profiles using many different SVAs. Here is the list of our publications that all are independent of the present study:
Kulski JK, Pfaff AL, Marney LD, Fröhlich A, Bubb VJ, Quinn JP, Koks S. Regulation of expression quantitative trait loci by SVA retrotransposons within the major histocompatibility complex. Exp Biol Med (Maywood). 2023 Dec;248(23):2304-2318. doi: 10.1177/15353702231209411. Epub 2023 Nov 30. PMID: 38031415; PMCID: PMC10903234.
Fröhlich A, Hughes LS, Middlehurst B, Pfaff AL, Bubb VJ, Koks S, Quinn JP. CRISPR deletion of a SINE-VNTR-Alu (SVA_67) retrotransposon demonstrates its ability to differentially modulate gene expression at the MAPT locus. Front Neurol. 2023 Sep 29;14:1273036. doi: 10.3389/fneur.2023.1273036. PMID: 37840928; PMCID: PMC10570551.
Fröhlich A, Pfaff AL, Bubb VJ, Koks S, Quinn JP. Characterisation of the Function of a SINE-VNTR-Alu Retrotransposon to Modulate Isoform Expression at the MAPT Locus. Front Mol Neurosci. 2022 Mar 9;15:815695. doi: 10.3389/fnmol.2022.815695. PMID: 35370538; PMCID: PMC8965460.
Koks S, Pfaff AL, Bubb VJ, Quinn JP. Expression Quantitative Trait Loci (eQTLs) Associated with Retrotransposons Demonstrate their Modulatory Effect on the Transcriptome. Int J Mol Sci. 2021 Jun 12;22(12):6319. doi: 10.3390/ijms22126319. PMID: 34204806; PMCID: PMC8231655.
Minor comments:
We have presented validation of these effects in our previous papers where we used alternative human cohorts (with 1,200 Parkinson’s disease patients), and we have also validated the effect of one particular SVA by applying CRISPR-mediated knockout of an SVA (SVA_67) that is located at the neurodegenerative risk locus, termed MAPT locus. In this study, we could demonstrate similar trends, highlighting this very SVA as a regulatory domain at the MAPT locus, and this is an independent cohort and validation. We have cited these articles. We have seen the same effect on SVAs in all these independent studies. Moreover, it appears that particular SVAs behave similarly in different studies. For example, SVAs that upregulate particular genes in one study cohort will also do it in different cohorts.
The genotyping accuracy of SVAs has also been verified in the lab by using PCR of the available samples, and the results have been published elsewhere (PMID: 34035310). In addition, we had 41 gDNA samples of individuals from a Parkinson’s cohort (PPMI) available to PCR validate bioinformatically determined polymorphisms for the presence/absence of a specific SVA of interest (SVA_67). All 41 genotypes identified by PCR, which determined the SVA_67 genotype as homozygous present (PP), heterozygous (PA) and homozygous absent (AA), respectively, matched with the bioinformatically generated genotypes by Delly, which validated not only the reliability of the results from our previous study but also within the cohort used in this study.
We have mentioned this now in the method section.
Hopefully, we were able to address all the comments and improve our manuscript.
Reviewer 2 Report
Comments and Suggestions for Authors
In this work, the authors used the public ALS data to investigate the SVA insertion Polymorphisms in shaping differential gene expression in the central nervous systems. They applied matrix expression quantitative trait loci analysis to evaluate the genetic loci (SVA elements) regulating genes. All the downstream analysis was based on matrix eQTL. The manuscript is easy to read and follow. Here are my concerns regarding this article.
Major:
(1) Matrix eQTL could be done by mutations (such as SNP, Indel, SV) other than SVAs. It would be very nice to see that the comparison between SVAs and other mutations on shaping differential gene expression.
(2)In lines 530-531, “For this analysis both ALS cases and healthy controls were combined. ” In this article, ALS cases and control were combined. Similar to subsection 2.4, did SVA RIPs display disease-specific gene expression?
Minor:
(3) Figure 1a(line 146) showed up later than Figure 1b(line 98) in the manuscript.
(4) What was the function of the top genes (such as FCGBP and MAPT) discovered by matrix eQTL? Were they related to nervous system?
Author Response
We thank all reviewers for the comments that helped us to improve and clarify our statements and results in the manuscript. Our responses are the following:
- This is an interesting point that can be considered in future experiments. However, this is currently out of the scope of this paper, where we specifically wanted to highlight the SVA insertion polymorphism-specific effect on differential gene expression. Future experiments can point out specific SNPs or other variations of interest (coding/non-coding variants) and correlate these with expression changes. However, an initial extraction of which mutations are of interest for disease pathways needs to be done to conceptualise follow-up studies.
-
In this study, the main goal was to highlight the general regulatory effect of SVAs in the brain. The available dataset is from the ALS consortium, which mostly contains ALS patients and a fraction of healthy controls. We pooled the cases and controls to get more general effects independent of the disease status. However, as our cohort is composed mainly of ALS patients and some of the gene patterns are similar to the known ALS patterns, we may be able to draw some conclusions regarding ALS. However, at this time, that does not allow us to classify that “SVA RIPs display disease-specific expression”. In addition, due to low n numbers in the ALS and control groups, individual group analysis was impossible. Therefore, future experiments investigating the influence of SVAs in ALS patients and control groups individually are required by using larger cohorts. We have mentioned this in the discussion.
-
We know that 1a follows 1b; however, for the generation of the figure and flow of the text, this was the best option as we started talking about the SVA structure followed by the project design. Hopefully, this is acceptable.
-
The MAPT gene and its protein product tau are crucial for various aspects of nervous system function, including microtubule stabilisation, axonal transport or synaptic plasticity. Mutations in the MAPT gene or dysregulation of tau protein expression and function are associated with several neurodegenerative diseases, collectively known as tauopathies. These diseases include Alzheimer's or frontotemporal dementia (compare backgrounds PMID: 20553310).
The gene encodes for the IgG Fc binding protein, FCGBP, which is crucial in immune protection and inflammation within the intestines. FCGBP is a vital intestinal mucosal immune defence component, contributing to anti-inflammatory processes and cellular protection. Its functions encompass aiding in immune responses and safeguarding the integrity of intestinal cells, highlighting its significance in maintaining gut health and regulating immune function (PMID: 36371440). The gut microbiota has been implicated in the pathogenesis of several neurological disorders, including amyotrophic lateral sclerosis (https://doi.org/10.1186/s12916-020-01885-3).
Hopefully, we were able to address all the comments and improve our manuscript.
Reviewer 3 Report
Comments and Suggestions for Authors
The manuscript referenced biomolecules-2898865 and titled “Exploring SVA Insertion Polymorphisms in Shaping Differential Gene Expression in the Central Nervous System” by Lauren S. Hughes and colleagues presents an interesting matrix expression quantitative trait loci study in whole genome sequencing and RNA sequencing data of amyotrophic lateral sclerosis patients (n=1544) and healthy controls (n=359) from the New York Genome Center ALS Consortium and demonstrated that reference SINE-VNTR-Alu (a TE) insertion polymorphisms can significantly modulate the expression of genes known to be involved in neurodegenerative diseases and especially in ALS. They observed that this was preferentially occurring in the trans position, in a tissue-specific manner and both in nuclear and mitochondrial genes. These findings along with previous ones obtained by these authors regarding Parkinson’s disease highlight the importance of studying the impact of TEs polymorphisms in the genomic landscape of neurodegenerative diseases, even more because the mobilization of TEs is a known phenomenon in the central nervous system cells, contributing both to hereditary cases and/or to acquired ones (de novo structural variants occurrence due to a number of factors, from environmental stresses to age events related to genome general derepression. In my opinion, the manuscript addresses a very important theme, it is sound, the data are robust and support the conclusions drawn by the authors and is well written, making it suitability for publication in Biomolecules journal after some minor corrections and discussion clarifications I list below.
Minor
- In my opinion, the introduction may be a bit too long.
- I would like the authors to explore the idea pointed in the Discussion, page 13, lines 424-428, namely “A potential mechanism for trans-acting eQTLs could be through the binding of CCCTC-binding factor (CTCF) to SVAs”. In the opinion of the authors, how can this binding occur?
- I’m not sure I understood the sentence in the Discussion, page 14, lines 465-466, namely “… SVA_30 and SVA_70 displayed the greatest regulatory effects on mitochondrial genes modulating five and four gene targets, respectively.”
- Discussion, page 15, lines 495-496: “Ultimately, the ability of SVAs to act as a regulatory domain could highlight the importance of TEs in the missing heritability of neurodegenerative disease.” Please rephrase. What do the authors really want to point out? The involvement of TEs in de novo cases? Is role as a regulatory domain generally? Both?
- Abstract, page 1, line 16, “…lateral sclerosis (ALS), however …”, delete “however”
- Discussion, page 14, lines 478-479: “… Since, the involvement of the immune response in neurological disease has been recognised the HLA locus has been…” delete “Since”
Comments on the Quality of English Language- Abstract, page 1, line 16, “…lateral sclerosis (ALS), however …”, delete “however”
- - Discussion, page 14, lines 478-479: “… Since, the involvement of the immune response in neurological disease has been recognised …” Please change the place of the comma to “… Since the involvement of the immune response in neurological disease has been recognized,…”
Author Response
We thank all reviewers for the comments that helped us to improve and clarify our statements and results in the manuscript. Our response is the following:
- For this study, we wanted to include a basic background on transposable elements and their potential involvement in disease so that the reader can follow the generated data and understand its importance.
-
It is known that transposable elements can affect gene function via binding transcription factors. It has been demonstrated that CTCF can bind the SVA VNTR in vitro, while the germline-expressed paralog CTCF-like can bind this domain in vivo. We have discussed this in the discussion and added that CTCF can bind the SVA (and cited this reference).
-
We have modified this sentence now. We have written the following:
"SVA_30 and SVA_70 displayed the greatest effect on mitochondrial genes, with SVA_30 modulating five targets and SVA_70 affecting four gene targets." -
We have modified the sentence with the following:
"Ultimately, the ability of SVAs to act as a regulatory domain can be an underappreciated source of genetic variation, and this may be involved in the mechanisms that lead to disease and disease progression. Here, differential gene expression caused by TE polymorphism could be one mechanism. " -
We have made all these changes in the text.
Round 2
Reviewer 1 Report
Comments and Suggestions for Authors
The authors have responded to all my comments. PCA did not identify stratification other than by tissue type, which is expected. The authors did not do hierarchical clustering of RNAseq data or WGS-derived PCA, these were intended as precautions against hidden subgroups that can be present in real-life data despite large n. Based on this work alone, there remains some uncertainty about their results - however, the results are consistent against their previous work. This lends credibility to their main results and their observations warrant publication.
Reviewer 2 Report
Comments and Suggestions for Authors I am happy with the revision and would like to congratulate the authors on the work.